

A functional tool to explore the reliability of micro-earthquake focal
mechanism solution for seismotectonic purposes
Guido Maria Adinolfi [1]*, Raffaella De Matteis [1], Rita De Nardis[2] and Aldo Zollo [3]
[1] Dipartimento di Scienze e Tecnologie, Università del Sannio Via dei Mulini, 59/A, 82100 Benevento,
Italy
[2] Dipartimento di Scienze Psicologiche, della Salute e del Territorio, Università di Chieti-Pescara "G.
d'Annunzio", Via dei Vestini, 32, 66100, Chieti, Italy
[3] Dipartimento di Fisica, Università di Napoli "Federico II", Complesso Universitario di Monte S.Angelo,
via Cinthia, 80124 Napoli, Italy
* Corresponding author: gmadinolfi@unisannio.it
ABSTRACT
Improving the knowledge of seismogenic faults requires the integration of geological, seismological,
and geophysical information. Among several analyses, the definition of earthquake focal mechanisms
plays an essential role in providing information about the geometry of individual faults and the stress
regime acting in a region. Fault plane solutions can be retrieved by several techniques operating in
specific magnitude ranges, both in the time and frequency domain and using different data.
For earthquakes of low magnitude, the limited number of available data and their uncertainties can
compromise the stability of fault plane solutions. In this work, we propose a useful methodology to
evaluate how well a seismic network used to monitor natural and/or induced micro-seismicity estimates
focal mechanisms as function of magnitude, location, and kinematics of seismic source and consequently
their reliability in defining seismotectonic models. To study the consistency of focal mechanism
solutions, we use a Bayesian approach that jointly inverts the P/S long-period spectral-level ratios and
the P polarities to infer the fault-plane solutions. We applied this methodology, by computing synthetic
data, to the local seismic network operated in the Campania-Lucania Apennines (Southern Italy) to
monitor the complex normal fault system activated during the Ms 6.9, 1980 earthquake. We
demonstrate that the method we propose can have a double purpose. It can be a valid tool to design
or to test the performance of local seismic networks and more generally it can be used to assign an
absolute uncertainty to focal mechanism solutions fundamental for seismotectonic studies.




## INTRODUCTION


Fault plane solutions represent a primary information that seismologists can retrieved to describe the
earthquake source.  The assessment of earthquake location, magnitude and focal mechanism reveals
the fundamental operations to characterize the earthquake source through the point source
approximation. After the earthquake location, origin time and dimension are identified, the focal
mechanism describes the basic geometry and kinematics of a point source in terms of strike, dip and
rake of fault plane along which the earthquake occurred. So, the focal mechanism is the most important
parameter that can be retrieved to recognize the geometry of the seismogenic faults, their style of
faulting and stress regimes active in a region. Moreover, the seismicity and focal mechanisms of events,
also of small magnitudes, are often used to constrain seismotectonic models, individual seismogenic
sources, the regional strain and stress fields. Consequently, an evaluation of their effective reliability
become a fundamental issue in seismotectonic studies.
Nevertheless, focal mechanisms cannot be calculated and constrained every time an earthquake occurs.
Although the calculation of focal mechanisms represents a routine analysis inside the seismological
agencies, the solutions are calculated only for a specific range of magnitude, usually greater than 4. In
fact, constraining the solution for earthquakes with small magnitude still represents a challenge, despite
the advancement in the technological process and the use of increasingly performing seismic networks.
This is due to several factors that we will analyse in detail. The techniques used to define the focal
mechanism of large-moderate earthquakes are based on the inversion of the moment tensor, that
corresponds to a stable and robust procedure, so much so that it is the most common method for this
type of analysis (Dreger, 2003; Delouis, 2014; Sokos and Zahradnik, 2013). This technique requires
an accurate knowledge of the propagation medium in relation to the range of frequencies used for the
modelling of the waveforms recorded during an earthquake. The smaller an earthquake, the higher the
frequency range of the signal to be modelled, the more detailed the knowledge and scale of the Earth's
interior must be.



Other analytical techniques are based on the recognition of radiation pattern that describes the
earthquake source. According to the position of seismic stations respect to the source, seismic waves
on seismograms show different amplitudes and polarities. These features are employed in a very simple
way by several algorithms to constrain the geometry of the earthquake faulting estimating the angular
parameters strike, dip and rake.  The classical method (Raesenberg and Oppenheimer, 1985;) uses the
P-wave polarities, but advanced ones use P- or S- wave amplitudes or amplitude ratios together with
first motions (Snoke, 2003) to better constrain the focal mechanism of small earthquakes. In fact, the
use of polarities alone is not convenient, especially if we consider the micro-seismicity (M < 3).  The
reasons could be the limited number of available data, their uncertainties and the difficulty to
automatically measure the  P-polarity with a sufficient degree of precision. For these reasons, different
techniques using different types of measurements such as P-wave amplitudes (Julian and Foulger, 1996;
Tarantino et al., 2019), P/S or S/P amplitude ratios measured in time or in the frequency domain
(Kisslinger et al., 1981; Rau et al., 1996; Hardebeck and Shearer, 2003; De Matteis et al., 2016),  or
S-wave polarizations (Zollo and  Bernard, 1991) have been developed. The joint inversion of polarities
and amplitude ratios led to more stable and robust solutions, allowing to account for geological site
effects and to decrease in first approximation the effects produced by the geometric and anelastic
attenuations.
Two kinds of errors generally influence the goodness of solution and retrieved model (Michele et al.,
2016). The perturbation errors that are related to how the uncertainty on data affect the model, and
the resolution errors refer to the capability to retrieve a correct model given a dataset as input. In other
words, how accurate could be the model that we can recover, even if error-free data are used. The sum
of perturbation and resolution errors corresponds to the final errors on the model obtained by solving
an inverse problem, as the solution of focal mechanism. In particular, the resolution errors depend on
the available data, and so on the initial condition of the inverse problem. In the case of focal mechanism,
number of seismic stations, as well as seismic network geometry, and velocity structure of the crust
influence the resolution and the reliability of the retrieved model.





How will the geometry of a seismic network determine the accuracy of focal mechanism solutions? The
answer to this question is not simple and requires a deep knowledge of geophysical and geological
characteristics of the region, often unrealistic. Moreover, the theoretical relationships that predict the
focal mechanism solutions for an earthquake scenario could be very complicated if several factors, such
as network configuration, noise level, source magnitude or source kinematics are taken into account.
We want to underline that a network configuration may be optimal for earthquake locations, but not for
retrieving fault plane solutions (Hardt and Scherbaum, 1994). In fact, its geometry may resolve some
fault kinematics better than others.
A seismic network layout is strictly associated to the goals of the network and to the available funds;
according to these features a network operator decides how many stations are required and where they
should be located (Havskov et al.; 2011). So, the number of seismic stations, the size and geometry of
network are defined after a preliminary phase based on the specific seismological target is evaluated
(Trnkoczy et al., 2009; Hardt and Scherbaum 1994; Steinberg et al. 1995; Bartal et al. 2000). The
number of seismic stations is related also to the dimension of the region to be monitored. Larger
regions require more stations, unless the seismicity target is represented only by strongest earthquakes.
In the case of small earthquakes, the available recordings come from only a portion of the total network,
while the distant stations show a seismic signal buried in the noise. In order to detect and locate low-
magnitude earthquakes, we must increase the number of seismic stations for area unit building a dense
seismic network. Beyond the number of stations that make up a seismic network, it is essential the
number of stations that properly record a seismic event, i.e. that provide not saturated seismic signals
with a high signal-to-noise ratio (Havskov et al.; 2011).
In this study, we propose a useful tool to evaluate both 1) the reliability of focal mechanism solutions
inferred by the inversion of different seismological data and 2) the performance of seismic network to
assess focal mechanism solutions and their errors. We evaluate the network capability to solve focal
mechanism as function of magnitude, location, and kinematics of seismic source.  We consider three
synthetic data measurements: P-wave polarities, P- S-wave amplitude spectral ratios and polarities and





amplitude ratios together. Moreover, different levels of noise are considered in order to simulate more
realistic conditions.
We selected as target the Irpinia Seismic Network (ISNet), a local seismic network that monitors the
Irpinia complex normal fault system (Southern Italy), activated during the Ms 6.9 earthquake of 23rd
November 1980. Evaluating the specific performance of an existing network for a seismological goal is
critical and can be used to decide how to improve its layout.

**METHOD**
We estimated focal mechanism using BISTROP code (De Matteis et al., 2016) that jointly inverts the
ratio between the P- and S-wave long-period spectral levels and the P-wave polarities according to a
Bayesian approach.  BISTROP has the advantage to use different observables for the determination of
fault plane solutions, such as the P/S long-period spectral level ratios or P-wave polarities, individually
or together. The benefits of the use of spectral level ratios are multiples: 1) they can be measured for a
broad range of magnitudes; 2) they can be calculated by automatic procedures without visual
inspection; 3) their estimates do not require to identify the first arrival time with  extreme precision, but
only a time window of signal containing P- or S-phase is mandatory and 4) as spectral amplitude ratios,
they can generally be used without the exact knowledge of the geological soil conditions (site effects)
and geometric/anelastic attenuation. Moreover, the joint inversion of amplitude spectral ratios and
polarities led to constrain fault plane solutions reducing the error associated to the estimates of
retrieved parameters. BISTROP solves an inverse problem through a probabilistic formulation leading
to a complete representation of uncertainty and correlation of the inferred parameters.
For a double couple seismic source, the radiation pattern depends on fault kinematics and relative
source-station position. In fact, it can be represented as function of 1) strike, dip and rake angles ($\varphi$, $\delta$,
$\lambda$) and 2) take-off and azimuth angles ($i_h$, $\varphi_r$). We can define the ratio between P- and S-wave radiation
pattern coefficients as:





$$\frac{\mathcal{R}^P\left(\phi, \delta, \lambda, i_h, \phi_R\right)}{\mathcal{R}^S\left(\phi, \delta, \lambda, i_h, \phi_R\right)} = \left(\frac{\alpha_s^2 \alpha_r}{\beta_s^2 \beta_r}\right) \ \frac{\Omega_0^P}{\Omega_0^S} \tag{1}$$


where $\Omega_0^P$ and $\Omega_0^S$ are the long-period spectral level of the P- and S-waves, respectively, and $\alpha_s$, $\alpha_r$,
$\beta_s$, $\beta_r$, are the P- and S-wave velocities at the source and at the receiver, respectively. The Equation 1
derives from the ratios of seismic moment for P- and S-waves defined as (Aki and Richards, 1980):

$$M_0 = \frac{4\pi \rho_s^{1/2} \rho_r^{1/2} \alpha_s^{5/2} \alpha_r^{1/2} R' \Omega_0^P}{F \langle|\mathcal{R}_{\theta\varphi}^P|\rangle} \tag{2}$$

and

$$M_0 = \frac{4\pi \rho_s^{1/2} \rho_r^{1/2} \beta_s^{5/2} \beta_r^{1/2} R' \Omega_0^S}{F \langle|\mathcal{R}_{\theta\varphi}^S|\rangle} \tag{3}$$


where $\rho_s$ and $\rho_r$ are the medium densities at the source and at the receiver, respectively; $\langle|\mathcal{R}_{\theta\varphi}^{SP}|\rangle$ and
$\langle|\mathcal{R}_{\theta\varphi}^S|\rangle$ are the average P- and S-wave radiation patterns, respectively. $R'$ is the geometrical spreading
estimated for a linear variation of velocity with depth (Ben-Menahem and Singh, 1981):
$$R' = \sqrt{\frac{\rho_r \alpha_r}{\rho_s \alpha_s}} \frac{R}{sini_h} \tag{4}$$

in which $R$ is the epicentral distance and $i_h$ is the take-off angle; $v_s$ and $v_r$ are the velocity at the source
and at the receiver to be substituted with $\alpha$ and $\beta$ values for the case of P- and S-wave. Thus, using
the displacement spectra, assuming a given source and attenuation model (Boatwright,1980), we can
derive from the signal recorded by a seismic station the ratio of radiation pattern coefficients for P- and





S-phases, as well as $\alpha$, $\beta$, $i_h$, $\varphi_r$ are known from the earthquake location and from the velocity model
used. So, from a theoretical point of view, the spectral amplitude ratios measured at several seismic
stations can be used to retrieve the ratio of radiation pattern coefficients $\mathcal{R}^P_{\theta\varphi}/\mathcal{R}^S_{\theta\varphi}$ as function of the
source-receiver azimuth and take-off angles.
BISTROP jointly inverts the spectral amplitude ratios with the observed P-wave polarities with the aim
of inferring the parameters $\varphi$, $\delta$, $\lambda$ of the focal mechanism in a Bayesian framework. A posterior
probability density function (PDF), for the vector of model parameter $\boldsymbol{m}$ ($\varphi$, $\delta$, $\lambda$) and the vector of
observed data $\boldsymbol{d}$, is defined as:
$$q(\boldsymbol{m}|\boldsymbol{d}) = \frac{f(\boldsymbol{d}|\boldsymbol{m})p(\boldsymbol{m})}{\int_M f(\boldsymbol{d}|\boldsymbol{m}')p(\boldsymbol{m}')\,d\boldsymbol{m}'} \qquad (5)$$


where $f(\mathbf{d}|\mathbf{m})$ is the conditional probability function that represents the PDF given the data $\mathbf{d}$ and for
parameter vector $\boldsymbol{m}$ in the model parameter space $\boldsymbol{M}$, and $p(\boldsymbol{m})$ is the a priori PDF. If P-wave polarities
and P/S spectral level ratios are independent datasets, the conditional probability function may be
written as:

$$f(\mathbf{d}|\mathbf{m}) = f(\boldsymbol{d}^L|\mathbf{m})f(\boldsymbol{d}^P|\mathbf{m}). \qquad (6)$$


in which the pdf of the data vector $\boldsymbol{d}^L$ of $N^L$ measurements of spectral ratios is multiplied for the pdf
of data vector $\boldsymbol{d}^P$ of $N^P$ measurements of P-wave polarities given the model $\boldsymbol{m}$.
Assuming that the observables have the same finite variance, for the $N^L$ observations of spectral level
ratios the conditional probability function may be defined as:




$$f(\boldsymbol{d}^L|\boldsymbol{m}) = \frac{1}{\left(\sqrt{2\pi}\sigma\right)^{N_L}} exp\left(-\frac{\sum_{i=1}^{N_L}\{d_i - [G(\boldsymbol{m})]_i\}^2}{2\sigma^2}\right)$$ (7)


Where $G(\boldsymbol{m})$ represents a functional relationship between model and data and corresponds to Equation
1 and $\sigma$ represents the uncertainty on the spectral measure.
For the $N^P$ observations of P-wave polarities, the conditional probability function is (Brillinger et al.,

189    1980):

$$f(\boldsymbol{d}^P|\boldsymbol{m}) = \prod_{i=1}^{N_P}\frac{1}{2}[1 + \psi(\mathcal{R}_i^P,\gamma_i,\rho_0)Y_i sign(\mathcal{R}_i^P)]$$ (8)


in which:

193    .

$$\psi(\mathcal{R}_i^P,\gamma_i,\rho_0) = (1 - 2\gamma_i)\,\text{erf}\left(\left|\rho_0\mathcal{R}_i^P(\boldsymbol{m})\right|\right)$$ (9)


The quantity reported in square brackets in Equation 8 represents the probability that the observed $i_{th}$
polarity $\gamma_i$ is consistent with the theoretical one computed from the model $\boldsymbol{m}$, whose theoretical P-wave
amplitude is  $\mathcal{R}_i^P$ and $sign(\mathcal{R}_i^P)$ is its polarity at $i_{th}$ station for a given fault plane solution. The
parameters $\rho_s$ and $\gamma_0$ , referring to the errors in ray tracing due to velocity model ambiguity and to the
uncertainty on polarity reading, regulate the shape of the PDF. For more details about the mathematical
formulation, see De Matteis et al., (2016).





**Irpinia Seismic Network**

Our analysis regards the area of the M 6.9, 1980 Irpinia earthquake (Southern Italy). Since 2005, ISNet, a local, dense seismic network monitors the seismicity along the Campania-Lucania Apennines covering an area of about $100 \times 70$ km$^2$ (Figure 1; Weber et al., 2007). The seismic stations are deployed within an elliptic area whose major axis, parallel to the Apennine chain, has a NW-SE trend with an average inter-stations distance of 15 km that reaches 10 km in the inner central zone. Each seismic station ensures a high dynamic range and it is equipped with a strong-motion accelerometer, Guralp CMG-5T or Kinemetrics Episensor, and a short period three-component seismometer, Geotech S13-J with natural period of 1 sec. In 6 cases, broadband seismometer is installed such as the Nanometrics Trillium with a flat response in the range 0.025–50 Hz. ISNet is operated by INFO (Irpinia Near Fault Observatory) and it provides real-time data at local control centres for earthquake early warning systems or real time seismic monitoring (Satriano et al., 2011). Seismic events are automatically identified and located from continuous recordings by automatic Earth-worm Binder and data are then manually revised by operators (Festa et al., 2020).

The 1980, M 6.9, Irpinia earthquake was one of the most destructive, instrumental earthquakes of the Southern Apennines, causing about 3000 fatalities and severe damages in the Campania and Basilicata regions. It activated a NW-SE trending normal fault system with a complex rupture process involving multiple fault segments according to (at least) three different nucleation episodes delayed each other of 20 s (Bernard and Zollo, 1989; Pantosti and Valensise; 1993; Amoruso et al.; 2005). No large earthquakes occurred in the Irpinia region since 1980. A Mw 4.9 earthquake took place in 1996 originating a seismic sequence inside the epicentral area of the 1980 earthquake (Figure 1; Cocco et al., 1999). Recent instrumental seismicity occurs mainly in the first 15 km of the crust showing fault plane solutions with normal and normal-strike slip kinematics, indicating a dominant SW-NE extensional regime (Pasquale et al., 2009; De Matteis et al., 2012; Bello et al.; 2021). Low-magnitude seismicity ($M_L < 3.6$) is spread into a large volume related to the activity of major fault segments of the 1980 Irpinia earthquake (Figure 1; Adinolfi et al.; 2019; Adinolfi et al.; 2020). Seismic sequences or swarms





often occurred in the area, extremely clustered in time (from several hours to few days) and space and
seem to be controlled by high pore fluid pressure of saturated Apulian carbonates bounded by normal
seismogenic faults (Stabile et al., 2012; Amoroso et al; 2014).

**DATA ANALYSIS**
We performed a resolution analysis of the reliability of focal mechanisms retrieved by data simulated at
ISNet. The analysis is carried out by evaluating the effect of 1) earthquake magnitude, 2) epicentral
location, 3) earthquake depth, 4) signal-to-noise ratio and 5) fault kinematics. We calculated the
capability of the local network to resolve fault plane solutions using different observables as input data:
a) P-wave polarities, b) P/S spectral amplitude ratios and c) polarities and amplitude ratios together.
In order to select   focal mechanisms to be used for our resolution study (Figure 2a), we carried out a
statistical analysis to define the most frequent fault plane solutions of instrumental seismicity.   We
classified according to the plunge of P- and T-axes the fault plane solutions reported in De Matteis et
al. (2012). The data refer to 2005-2011 seismicity occurred in the Irpinia, excluding those occurred in
Potenza region.   As shown in Figure 2b, splitting the range of the data into equal-sized bins, we
selected the focal mechanism corresponding to the most populated class. We report it in Figure 2a as
FM2. This corresponds to normal-strike-slip fault plane solution with strike, dip and rake equal to 292°,
53° and -133°, respectively. Then, we decide to test the focal mechanism solution of the1980 Irpinia
earthquake, a pure normal fault (strike, dip, rake: 317°, 59°, -85°; Westaway and Jackson; 1987; Fig.
2a) here and after FM1. This solution is very similar to the focal mechanism corresponding to: 1) the
regional stress field (see Supplementary Material); 2) the $M_L$ 2.9, Laviano earthquake, one of the most
energetic earthquakes of the last years (Stabile et al.; 2012), and 3) those of the 2nd, 3rd, 4th bins. Finally,
we selected the solution corresponding to the 5th bin reported as FM3 in Figure 2a. This focal
mechanism is quite different from the others due to a predominant component along the fault strike
(strike, dip, rake: 274°, 71°, -128°)





For each of three selected fault plane kinematics, we calculated synthetic data (P-wave polarities or P- and S-wave spectral amplitudes) at seismic stations varying the earthquake location and by using a local velocity model (Matrullo et al., 2013). We discretize the study area with a square grid (100 X 100 $km^2$), centred on the barycentre of ISNet, with 441 nodes and sampling step of 5 km. Each node corresponds to a possible earthquake epicentre (Figure 3).

For each grid node and according to the earthquake magnitude to be tested, we have to select the ISNet stations for simulations. The number of seismic stations that record an event depends on earthquake magnitude, source-stations distance, crustal medium properties and on the level of noise. Theoretical relationships that link the seismic source to the signal recorded at each single station are quite complicated and are based on the accurate knowledge of crustal volume in which the seismic waves propagated, such as the three-dimensional wave velocity structure, anelastic attenuation or site conditions of single receiver. In order to overcome this limitation, we used an empirical relationship to define the number and the distance of the seismic stations that record a seismic signal as function of magnitude, once its epicentral location (grid node) and depth are fixed. Using the bulletin data retrieved by INFO at ISNet during the last two years (January 2019-March 2021; http://isnet-bulletin.fisica.unina.it/cgi-bin/isnet-events/isnet.cgi), we selected two earthquake catalog datasets with depth equal to 5 (+- 2) km and 10 (+- 2) km, respectively and local magnitude ranging between 1.0 and 2.5. These choices are motivated by the characteristics of the Irpinia micro-seismicity recorded by ISNet. Then, we divided each dataset in bins of 0.5 magnitude and for each bin we retrieved the median number of P-wave polarity readings and the median epicentral distance of the farthest station that recorded the earthquake (Table 1). The bulletin data are manually revised by operators, and we selected only seismic records that provide P- and/or S- wave arrival times. The median value of the distance of the farthest station is then used to select the seismic stations for which synthetic data are calculated. We run simulations only for earthquake recorded at least by 6 seismic stations. The synthetic P-wave polarities are simulated only at a number of stations corresponding to the median value previously defined. (Table 1). We pointed out that the number of P-wave polarities empirically assigned is related





to the available earthquake catalogue data of the Irpinia region where the seismicity can occur in
different portions of area covered by the network, not always with an optimal azimuthal coverage.
Additionally, we simulated the uncertainty on the measure of spectral level ratios or the effect of seismic
noise adding a gaussian noise to the synthetic data with two different percentage levels, as 5% and

286 30%.


With this configuration, we simulated:

• Three datasets of seismic observables: P-wave polarities (D1), P/S spectral level ratios (D2) and
polarities and P/S spectral level ratios together (D3)
• Two hypocentre depths: 5 km and 10 km
• Three magnitude bins: $M_L$ 1.0 -1.5 (M1), $M_L$ 1.5 - 2.0 (M2) and $M_L$ 2.0 - 2.5 (M3)
• Three fault plane solutions: FM1 (317°, 59°, -85°), FM2 (292°, 53°, -133°) and FM3 (274°, 71°,
-128°)
• Two level of gaussian noise: 5% and 30%


**DISCUSSION**
In order to evaluate the seismic network capability to resolve fault plane solutions, we defined five kinds
of map to study how the focal mechanism (FM) resolution and error spatially change in the area where
ISNet is installed:

• Kagan angle misfit map (KAM)
• Map of the focal mechanism parameter misfit (FMM)
• Strike, Dip and Rake error map (FME)
• Kagan angle average map (KAA)



•   Kagan angle standard deviation map (KAS)

The Kagan Angle (KA) measures the difference between the orientations of two seismic moment tensors
or two double couples. It is the smallest angle needed to rotate the principal axes of one moment tensor
to the corresponding principal axes of the other (Kagan et al.; 1991; Tape and Tape; 2012). The smaller
the KA between two focal mechanisms, more similar they are. In KAM map, for each node the value of
KA between the theoretical and retrieved solution is reported, while in FMM map, the absolute value of
the misfit between the strike, dip and rake angles of the retrieved and theoretical solution is indicated.
FME is defined as the error map of strike, dip and rake in which the uncertainties (standard deviations)
are calculated considering all the solutions with probability larger than the 90% (S90) of the maximum
probability, corresponding to the best solution retrieved. Additionally, these solutions are used to study
how constrained is the FM solution. In fact, the KA is calculated between each FM of S90 solutions and
the retrieved best solution. The mean and the standard deviation of resulting KA distribution are plotted
in KAA and KAS maps, respectively. The smaller KA mean and std, the more constrained is the obtained
fault plane solution.
We consider the FM1, i.e. the focal mechanism of the 1980 Irpinia earthquake located at 10 km depth,
first. Looking at Figures 4 and 5, we see the effect of using the three different datasets. Considering
D1, we can calculate the FM only for earthquakes with magnitude 2.0-2.5 for which at least 6 polarities
are available. As shown by KAM map in Figure 4a, the retrieved solutions are characterized by high KA
(> 50°) with limited areas or single nodes with values in the range 40°-50°. Therefore, D1 is not
sufficient to retrieve with acceptable accuracy the FMs for earthquakes with magnitude 2.0-2.5. The
same result is obtained for FM2 and FM3 (Figure 4b-c). Comparing the results of the simulations using
D2 and D3 (Figure 5), the accuracy of retrieved solution is improved when P-wave polarities data are
added to spectral level ratios. In fact, the areas in KAM map with high value of KA (KA $\geq$ 18°; red or
green areas) disappear or are strongly reduced. Nevertheless, we want to underline that, even with D2
dataset, except in some small areas, the FMs are well retrieved for all magnitudes with the KA misfit
mostly lesser than 10°. The spatial resolution of the network is strongly influenced by the earthquake



magnitude. In fact, for both M1 and M2, there are nodes (white areas with KA = -1) for which the FMs
cannot be calculated because a minimum number of stations (at least 6) are not available (Table 1). At
the same time, the areas better resolved correspond to the region inside the network, although with
D2 and D3 acceptable solutions are calculated for M1 and M2 earthquakes also outside the network,
(Figure 5).
Looking at Figure 6, using the D3 dataset, we observe that, among the FM parameters, the dip angle
is the best resolved compared with strike and rake angles. Considering M2 an M3 focal mechanisms,
the misfit of dip is very low (< 8°), followed, in ascending order, by rake and strike that show higher
values ( 10° < misfit < 22°). For M1 (Figure 6a-d-g), rake and strike misfits are larger than 50°, with
rake worse resolved than strike. The unresolved areas correspond to the regions outside the seismic
network.
The KAA and KAS maps (Figures 7 and 8) show how the network constrains the fault plane solution as
function of the epicentral location. Moreover, the Figures 7d-e-f and 8d-e-f indicate that the areas with
KA mean and std greater than 30° and 20°, respectively, are reduced when P-wave polarities and
spectral level ratios data are used. On contrary, only for M1 focal mechanisms there is no improvement
because the number of P-wave polarities is the same for both D2 and D3 datasets (Table 1). The worst
constrained regions correspond to a belt surrounding the seismic network, with KA mean < 30° and
KA std < 20° for M2 and M3 solutions. For M1, areas with high uncertainty   remain outside and inside
the network, specifically in central and in the eastern sectors.
Looking at the uncertainties of FM parameters, obtained by using the D3 dataset, the Figure 9 shows
that the dip is the better constrained parameter with an error < 10°, also for M1 solutions. The rake
angle shows an uncertainty lesser than 20° for M2 and M3, while it overcomes 50° for M1. The strike
angle reveals the highest uncertainty, with values greater than 50° in the eastern and southern sectors
of the map for all analysed magnitudes (M1, M2 and M3). Accuracy improves moving from M1 to M3
earthquakes.
As shown in Figure 10, the accuracy of fault plane solutions, in terms of KA misfit calculated by using
the D3 dataset, is similar for FM1, FM2 and FM3, mostly with values lesser than 8° for all the



magnitudes M1, M2 and M3. FM2 and FM3 show a slightly higher precision than FM1 in the area inside
the seismic network (see FMM, FME, KAA and KAS maps for FM2 and FM3 in Supplementary Material).
In the regions outside the network, where the azimuthal gap increases, the FMs better constrained in
descending order are: FM3, FM2 and FM1. This effect should be due to geometric relationship between
the spatial distribution of the seismic stations and the orientation of the principal axes (P, T, B) that
characterize the FMs.
Considering the effect of hypocentre depth, the results achieved for earthquakes at 5 km depth, by
using the D3 dataset, are overall unchanged (Figure 11).  We note that the fault plane solutions are
slightly worse resolved due to a smaller number of P-wave polarities available for M2 and M3. The KA
misfit mainly is lesser than 10°, even though the number and the dimension of areas with misfit > 20°
are greater than those obtained considering earthquakes at 10 km depth. Moreover, the dip angle
shows a misfit lower than strike and rake angles for M1, M2 and M3; the accuracy of the retrieved FMs
parameters is mainly lesser than 8°, as shown in Figure 11.
Previous analyses are carried out considering by using data affected by 5% gaussian error.  In the last
test, we simulated synthetic data adding a 30% gaussian error. As illustrated in Figure 12, FM solutions
show overall larger misfit, in particular   the KA   inside the seismic network is lesser than 20°. The area
best resolved (KA < 8°) is considerably y reduced to a small central portion of the network. This result
indicates that the accuracy of the spectral level ratio estimates is crucial: noisy waveforms with low
signal to noise ratio can critically affect the result of the focal mechanism inversion. So, seismic noise
as well as the number of available stations, variable due to the operational condition, strongly influence
the capability of the seismic network to retrieve fault plane solution. Using the results of our simulations,
we classified the focal mechanism provided by De Matteis et al., (2016) according to a quality code
based on the resolution of fault kinematics (Table 2). In fact, we assigned to focal mechanisms of Irpinia
instrumental seismicity a quality A, B and C for solutions corresponding to FM3, FM2 and FM1
kinematics, respectively. The quality A, B and C correspond to the average value of KA misfit (FM1=4.5°,
FM2=3.1°, FM3=2.4°) calculated for M1, M2 and M3 magnitudes using D3 dataset and considering
earthquakes at 10 km depth with 5% gaussian errors.




### CONCLUSIONS

We studied the focal mechanism reliability retrieved by the inversion of data recorded by ISNet, a local
dense seismic network that monitors the Irpinia Fault System in Southern Italy. Three different datasets
of seismological observables are used as input data for focal mechanism determination: a) P-wave
polarities, b) P/S spectral amplitude ratios and c) joint polarities and amplitude ratios. Starting from
empirical observations, we computed synthetic data for a regular grid of epicentre locations at two
depths (5 and 10 km), for earthquake magnitude in the range 1.0-2.5 and for three focal mechanism
solutions.  Two different levels of  gaussian error (5% and 30%) are added to the data.
Main conclusions can be summarized as follows.
• The joint inversion of P-wave polarities and P/S spectral amplitude ratios allows to retrieve
accurate FM (KA misfit < 8°) also for earthquakes with magnitude ranging between 1.0 and 2.5,
at depth of 5 and 10 km. Due to the low-energy magnitude, the number of P-wave polarities is
not adequate to constrain fault plane solutions.
• The spatial resolution analysis of ISNet shows that the most accurate FM solutions are obtained
for earthquakes located inside the network with    strike, dip and rake misfit < 8°. Nevertheless,
outside the network or at its borders, acceptable solutions can be calculated even if the
azimuthal coverage is not adequate (especially for M2 and M3 events). This peculiarity is due to
geometrical relationship between the recording seismic stations and the orientation of principal
axes (P, T, B).
• The geometry of the network allows to well resolve fault plane solutions varying between normal
and normal-strike focal mechanism with mainly strike, dip and rake misfit lesser than 10° for
magnitude range 1.5-2.5. The network resolves slightly better normal-strike fault plane solution
than a pure normal focal mechanism.
• Among the FM parameters, the dip angle shows the lowest uncertainty. Strike and rake angles
reveal higher errors especially for M 1-1.5 earthquakes in the region outside the seismic network.




• Dataset affected by 30% gaussian error provide a worsening in the accuracy of the retrieved

416        FMs. Although the high error level, the area of well resolved fault plane solutions (KA misfit <

417        20°) persists in the central part of the network, especially for M2 and M3.

The methodology described in this work can be a valid tool to design or to test the performance of
local seismic networks, operated to monitor natural or induced seismicity. Moreover, given a network
configuration, it can be used to evaluate the reliability of FMs that represent a fundamental information
in seismotectonic studies. Although it is a theoretical study, many earthquake scenarios with several
magnitude, locations and noise conditions can be simulated to mimic the real seismicity.





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





**TABLES**

| Depth 5 km | Max Distance (km) | No. P-polarities |
|---|---|---|
| $M_L$ 1.0 -1.5 | 30 | 1 |
| $M_L$ 1.5 - 2.0 | 49 | 1 |
| $M_L$ 2.0 - 2.5 | 57 | 4 |
| Depth 10 km | Max Distance (km) | No. P-polarities |
| $M_L$ 1.0 -1.5 | 33 | 1 |
| $M_L$ 1.5 - 2.0 | 40 | 5 |
| $M_L$ 2.0 - 2.5 | 66 | 6 |

**Table 1** Maximum distance of the farthest triggered seismic station and number of P-wave polarities as function of earthquake magnitude and depth. The relations, empirically derived, are used for the earthquake simulations.





| P-plunge (°) | P-trend (°) | T-plunge (°) | T-trend (°) | Strike (°) | Dip (°) | Rake (°) | Quality |
|---|---|---|---|---|---|---|---|
| 55 | 344 | 31 | 196 | 325 | 20 | -40 | A |
| 51 | 334 | 36 | 181 | 320 | 15 | -30 | A |
| 55 | 14 | 31 | 226 | 355 | 20 | -40 | A |
| 53 | 205 | 34 | 49 | 180 | 15 | -40 | A |
| 55 | 72 | 33 | 272 | 35 | 15 | -50 | A |
| 51 | 177 | 32 | 37 | 290 | 80 | -110 | A |
| 54 | 292 | 34 | 91 | 10 | 80 | -80 | A |
| 77 | 146 | 9 | 7 | 270 | 55 | -100 | B |
| 80 | 235 | 10 | 55 | 325 | 55 | -90 | B |
| 76 | 103 | 2 | 6 | 110 | 45 | -70 | B |
| 76 | 117 | 2 | 214 | 290 | 45 | -110 | B |
| 76 | 82 | 7 | 199 | 275 | 40 | -110 | B |
| 75 | 190 | 15 | 10 | 280 | 60 | -90 | B |
| 75 | 205 | 15 | 25 | 295 | 60 | -90 | B |
| 85 | 230 | 5 | 50 | 140 | 40 | -90 | B |
| 83 | 146 | 0 | 53 | 150 | 45 | -80 | B |
| 80 | 240 | 10 | 60 | 330 | 55 | -90 | B |
| 81 | 233 | 5 | 353 | 270 | 50 | -80 | B |
| 81 | 347 | 5 | 227 | 130 | 50 | -100 | B |
| 55 | 93 | 10 | 198 | 255 | 45 | -140 | C |
| 55 | 133 | 10 | 238 | 295 | 45 | -140 | C |
| 48 | 130 | 2 | 38 | 275 | 60 | -140 | C |
| 48 | 305 | 2 | 37 | 340 | 60 | -40 | C |
| 55 | 202 | 7 | 102 | 345 | 60 | -130 | C |
| 58 | 121 | 2 | 27 | 270 | 55 | -130 | C |
| 58 | 131 | 2 | 37 | 280 | 55 | -130 | C |
| 55 | 342 | 7 | 242 | 125 | 60 | -130 | C |
| 47 | 138 | 11 | 36 | 165 | 50 | -30 | C |
| 49 | 182 | 14 | 289 | 340 | 45 | -150 | C |
| 58 | 151 | 2 | 57 | 300 | 55 | -130 | C |
| 49 | 168 | 14 | 61 | 190 | 45 | -30 | C |
| 59 | 308 | 15 | 64 | 355 | 65 | -60 | C |
| 57 | 306 | 14 | 59 | 115 | 40 | -140 | C |
| 57 | 76 | 14 | 189 | 245 | 40 | -140 | C |
| 45 | 85 | 6 | 348 | 225 | 65 | -140 | C |
| 55 | 22 | 7 | 282 | 165 | 60 | -130 | C |
| 57 | 241 | 14 | 354 | 50 | 40 | -140 | C |
| 55 | 98 | 7 | 198 | 135 | 60 | -50 | C |
| 51 | 115 | 2 | 22 | 145 | 55 | -40 | C |
| 55 | 147 | 7 | 47 | 290 | 60 | -130 | C |

**Table 2.** Fault plane solutions of instrumental seismicity occurred in Irpinia region in 2005-2008 and
calculated by De Matteis et al., (2012). The solutions are classified according to a quality code based
on the resolution of fault plane kinematics as derived in this study. The result of our simulations
suggests a quality as follows: FM1=C, FM2=B, FM3=A.



**FIGURES**

**Figure 1.** Epicentral map of the earthquakes (green circles) recorded by Irpinia Seismic Network (ISNet,
red triangles) from 2008 to 2020 (http://isnet-bulletin.fisica.unina.it/cgi-bin/isnet-events/isnet.cgi). The
yellow and orange stars refer to the epicentral location of the 1980, M 6.9, and of the 1996, M 4.9
earthquakes, respectively. Historical seismicity is shown with black squares (I0 ≥ 6–7 MCS). Seismogenic
sources related to the Irpinia fault system are indicated by orange rectangles; potential sources for
earthquakes larger than M 5.5 in surrounding areas are indicated in grey (Database of Individual
Seismogenic Sources, DISS, Version 3.2.1)



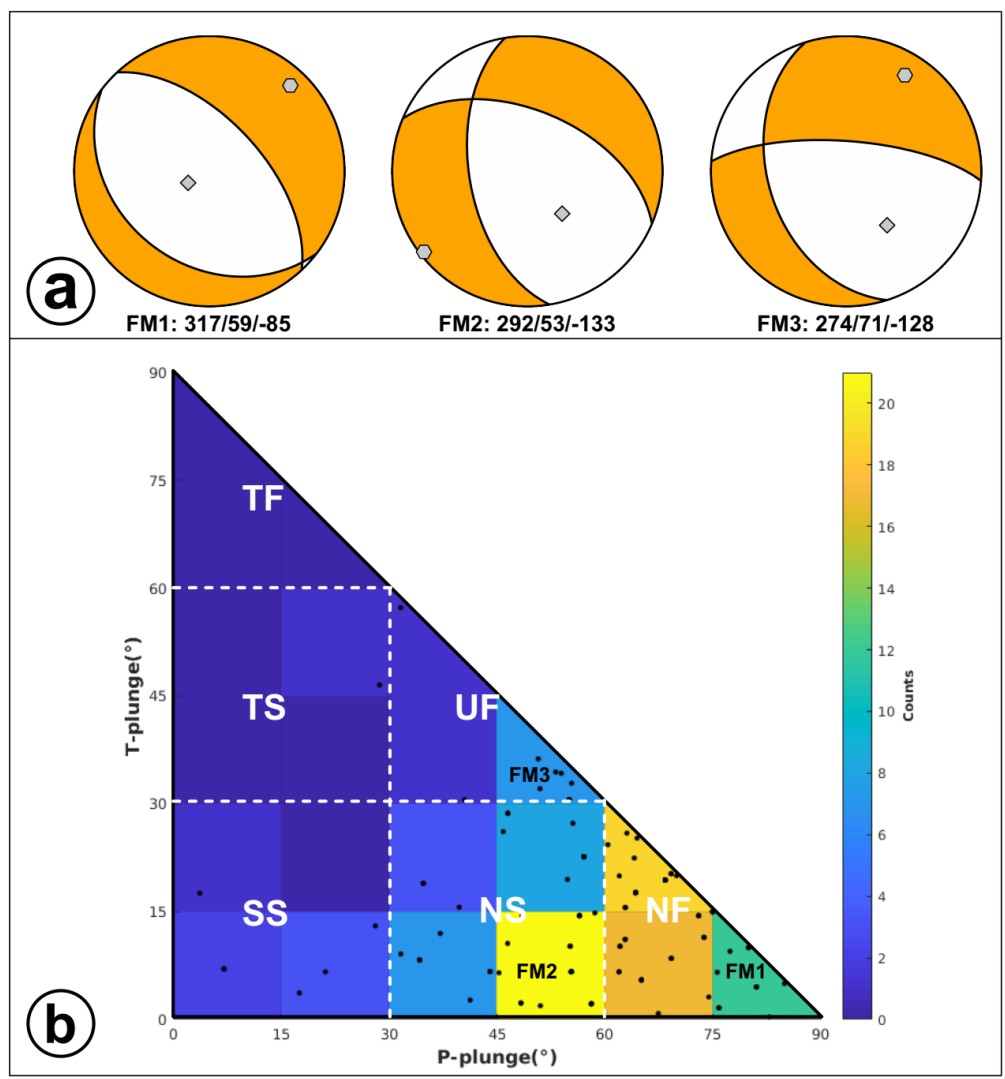

**Figure 2.** Fault plane solutions used for earthquake simulations. a) From left to right: 1) Ms 6.9, 23rd November 1980 (FM1; Westaway ) 2) and 3) Median focal mechanism found from solutions from the 1st (FM2) and 5th (FM3) most populated bin of histogram of panel b. b) Fault plane solution classification according to the plunge of P- and T-axes with specific tectonic regimes (Legend: NF, normal fault; NS, normal-srike; SS, strike-slip; TF, thrust ; TS, thrust-strike; UF, unknown fault). The number of earthquakes (color bar) is counted in bins of 15° × 15°.



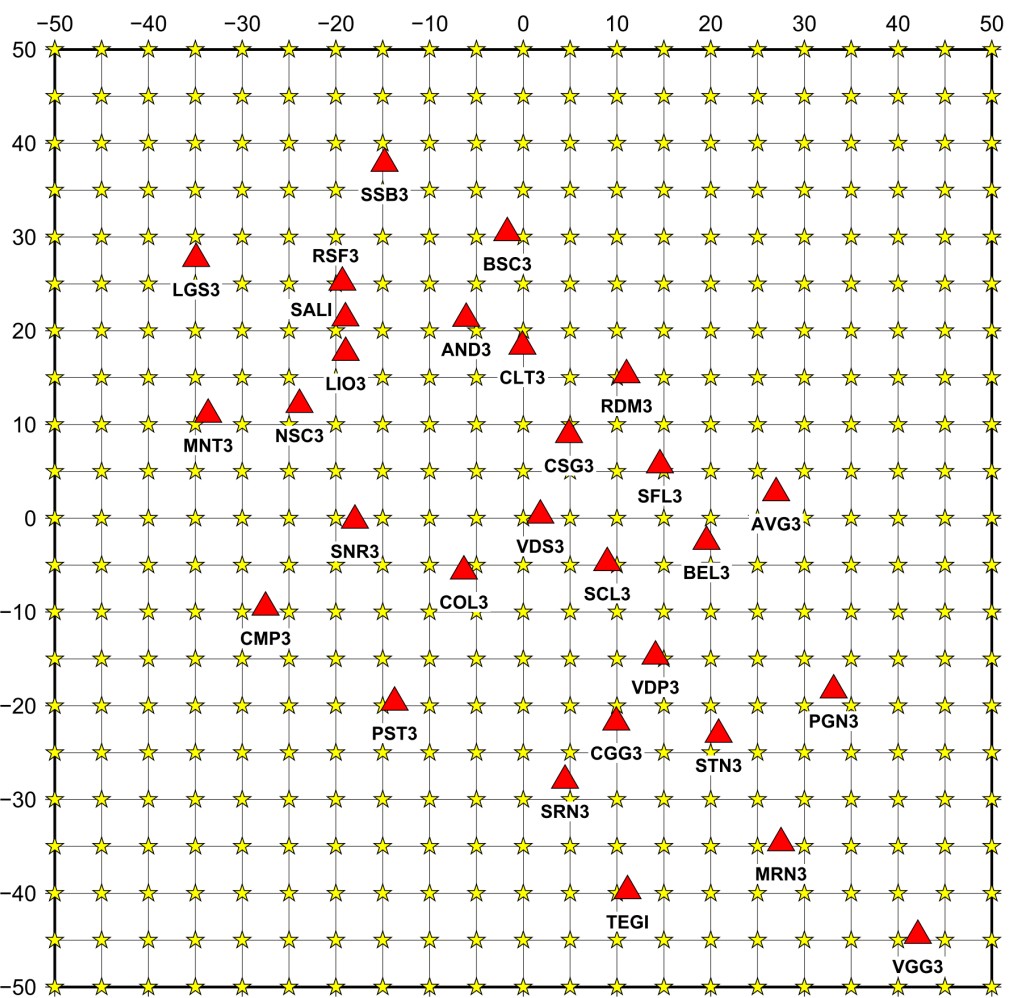

**Figure 3.** Regular grid of epicentres (yellow stars) used for simulating earthquakes. The area is 100x100 km² with 5 km of spacing along both horizontal coordinates. Irpinia Seismic Network (ISNet) is reported with red triangles.





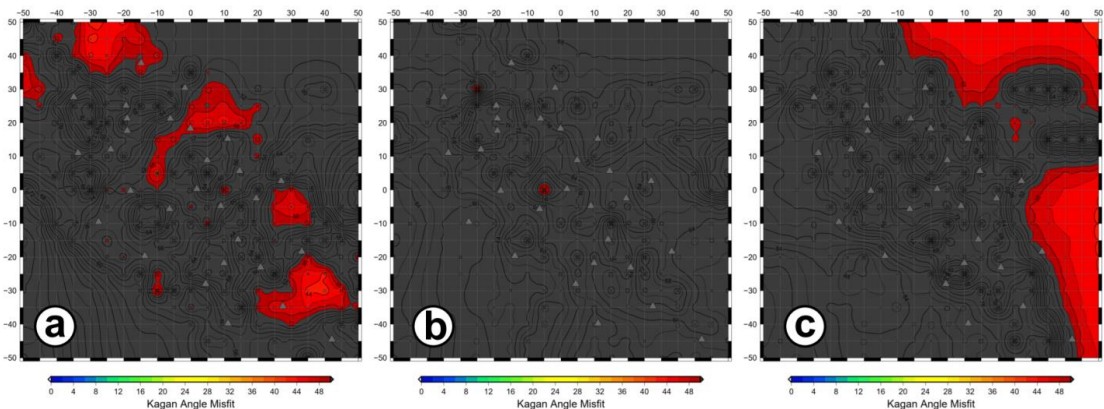

**Figure 4.** KAM (Kagan angle misfit) map for retrieved focal mechanisms with D1 dataset as input data and simulating earthquakes with M3 magnitude and FM1 (a), FM2 (b) and FM3 (c) theoretical fault plane solution at 10 km depth.



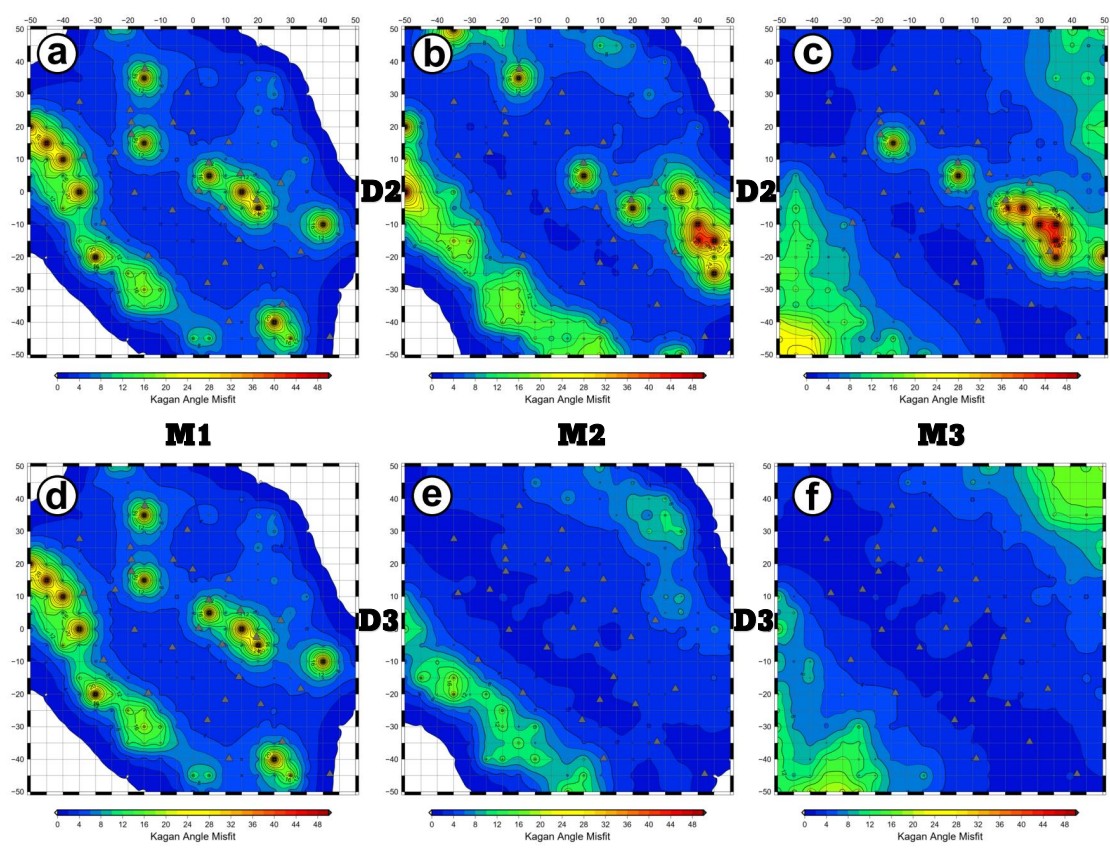

**Figure 5.** KAM (Kagan angle misfit) map for retrieved focal mechanisms with D2 (a, b, c) and D3 (d, e,
f) datasets as input data and simulating earthquakes with M1 (a, d), M2 (b, e) and M3 (c, f) magnitudes
and FM1 theoretical fault plane solution at 10 km depth. The level of gaussian noise is set to 5%.





**Figure 6.** FMM (focal mechanism parameter misfit) maps for retrieved focal mechanisms with D3 datasets as input data and simulating earthquakes with M1 (a, d, g), M2 (b, e, h) and M3 (c, f, i)



magnitudes and FM1 theoretical fault plane solution at 10 km depth. a, b, c refer to strike misfit; d, e,
f refer to dip misfit; g, h, i refer to rake. The level of gaussian noise is set to 5%.



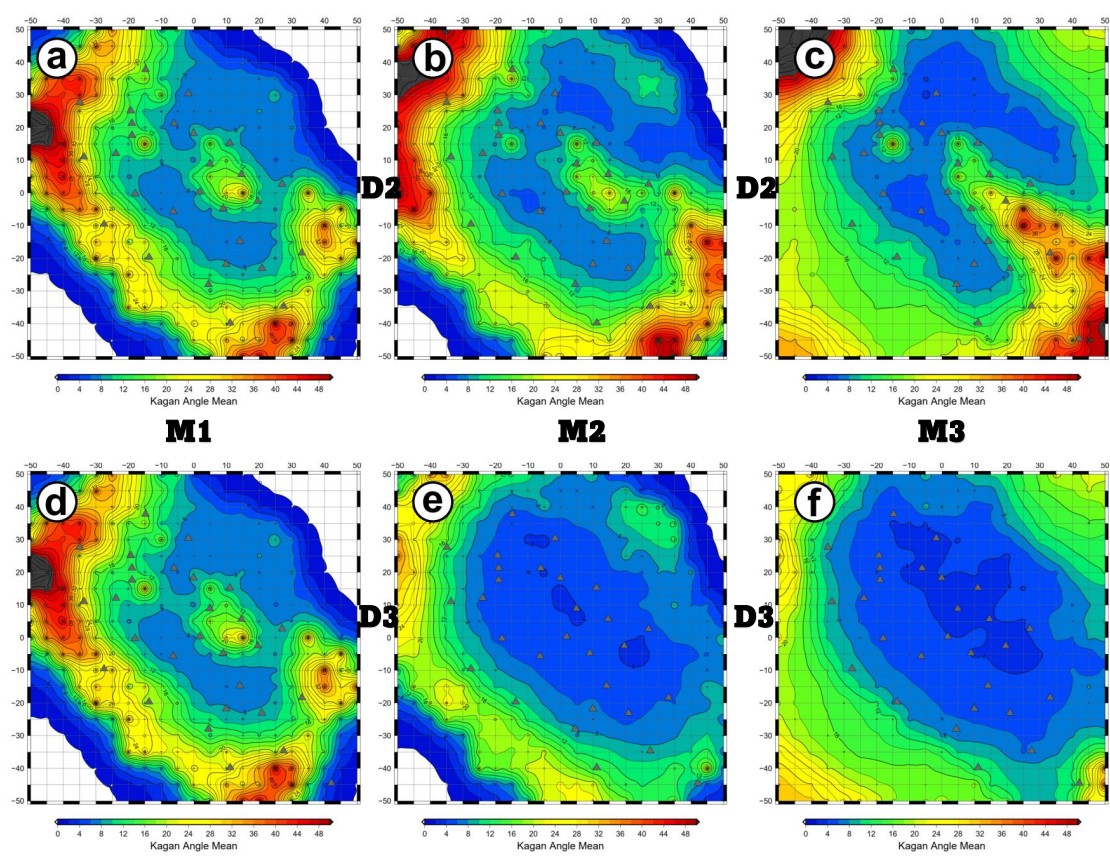

**Figure 7.** KAA (Kagan angle average) maps for retrieved focal mechanisms with D2 (a, b, c) and D3 (d,
e, f) datasets as input data and simulating earthquakes with M1 (a, d), M2 (b, e) and M3 (c, f) magnitudes
and FM1 theoretical fault plane solution at 10 km depth. The level of gaussian noise is set to 5%.

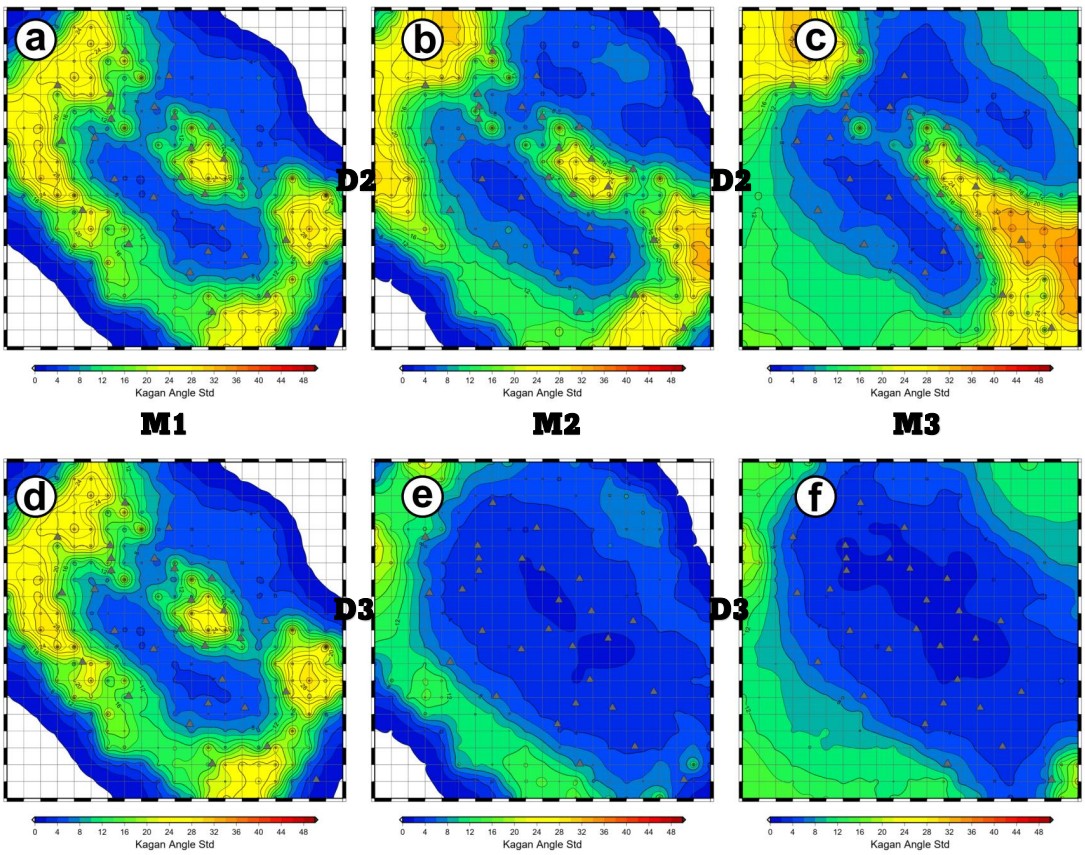

**Figure 8.** KAS (Kagan angle standard deviation) maps for retrieved focal mechanisms with D2 (a, b, c)
and D3 (d, e, f) datasets as input data and simulating earthquakes with M1 (a, d), M2 (b, e) and M3 (c,
f) magnitudes and FM1 theoretical fault plane solution at 10 km depth. The level of gaussian noise is
set to 5%.





**Figure 9.** FME (strike, dip and dake error) maps for retrieved focal mechanisms with D3 datasets as input data and simulating earthquakes with M1 (a, d, g), M2 (b, e, h) and M3 (c, f, i) magnitudes and





FM1 theoretical fault plane solution at 10 km depth. a, b, c refer to strike error; d, e, f refer to dip error;
g, h, i refer to rake error. The level of gaussian noise is set to 5%.





Figure 10. KAM (Kagan angle misfit) maps for retrieved focal mechanisms with D3 datasets as input
data and simulating earthquakes with M1 (a, d, g), M2 (b, e, h) and M3 (c, f, i) magnitudes and FM1 (a,



b, c), FM2 (d, e, f) and    FM3 (g, h, i) theoretical fault plane solution at 10 km depth. The level of
gaussian noise is set to 5%.






**Figure 11.** FMM (focal mechanism parameter misfit) maps for retrieved focal mechanisms with D3
datasets as input data and simulating earthquakes with M1 (a, d, g), M2 (b, e, h) and M3 (c, f, i)





magnitudes and FM1 theoretical fault plane solution at 5 km depth. a, b, c refer to strike misfit; d, e, f
refer to dip misfit; g, h, i refer to rake. The level of gaussian noise is set to 5%.

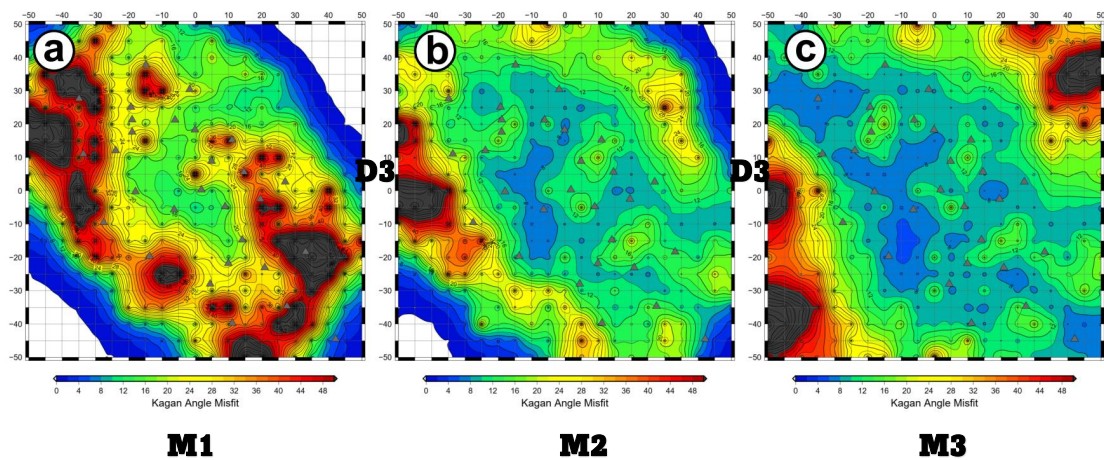

**Figure 12.** KAM (Kagan angle misfit) map for retrieved focal mechanisms with D3 (a, b, c) datasets as input data and simulating earthquakes with M1 (a), M2 (b) and M3 (c) magnitudes and FM1 theoretical fault plane solution at 10 km depth. The level of gaussian noise is set to 30%.