# Peer review of "A functional tool to explore the reliability of micro-earthquake focal mechanism solution for seismotectonic purposes"

_Solid Earth, 2021_

## Referee Comment (RC1)

A functional tool to explore the reliability of micro-earthquake focal mechanism solution for seismotectonic purposes
by Adinolfi G. M., De Matteis R., De Nardis R., and Zollo, A.
posted on the Solid Earth Discussions (SED)

**General comments**
The work provides a valid tool for testing the reliability of seismic networks in determining fault plane solutions. This is profitable in seismotectonic studies and is strictly relevant to the goals of the Solid Earth special issue. The proposed method has been applied to micro-earthquakes recorded by the ISNet seismic network in Southern Italy in the period 2005-2011. The authors used a Bayesian approach that jointly inverts the P/S long-period spectral-level ratios and the P polarities to infer the fault-plane solutions. They also describe an application to the ISNet catalogue.
The work is well presented, the subdivision into paragraphs is appropriate, the methodology is clearly stated, as well as results and conclusions. I suggest minor revisions.

**Specific comments**
Lines 267-269: "*In order to overcome this limitation, we used an empirical relationship to define the number and the distance of the seismic stations that record a seismic signal as function of magnitude, once its epicentral location (grid node) and depth are fixed.*"
If the authors used an empirical relationship, it should be given. Actually, later in the text they state that they chose for each magnitude bin the median values of the distance of the farthest triggered seismic station and of the number of P-wave polarities. Please, clarify this point.

Lines 382-388: "*Using the results of our simulations, we classified the focal mechanism provided by De Matteis et al., (2016) according to a quality code based on the resolution of fault kinematics (Table 2). In fact, we assigned to focal mechanisms of Irpinia instrumental seismicity a quality A, B and C for solutions corresponding to FM3, FM2 and FM1 kinematics, respectively. The quality A, B and C correspond to the average value of KA misfit (FM1=4.5° FM2=3.1°, FM3=2.4°) calculated for M1, M2 and M3 magnitudes using D3 dataset and considering earthquakes at 10 km depth with 5% gaussian errors.*"
The authors indicated multiple factors that influence the goodness of the fault plane solution, such as the magnitude, the observables from waveforms, network geometry, the noise level. They should point out that this is an example of application of the method, and that the proposed classification only concerns the type of mechanism. I would suggest giving more emphasis to this paragraph.

**Technical corrections**
Lines 29-30: "*We applied this methodology, by computing synthetic data, to the local seismic network operated in the Campania-Lucania Apennines*"
I suggest changing to: "We applied this methodology, by computing synthetic data, to the local seismic network **operating** in the Campania-Lucania Apennines"

Line 46: "*After the earthquake location, origin time and dimension are identified*"
I suggest changing to:
"After the earthquake location, origin time and dimension **source** are identified"

Line 53: Please, change "*become*" to "**becomes**".

Line 61: Please, change "*so much so that*" to "**so much that**".

Lines 69-71: "*These features are employed in a very simple way by several algorithms to constrain the geometry of the earthquake faulting estimating the angular parameters strike, dip and rake*"
I suggest changing to:
"These features are employed in a very simple way by several algorithms to constrain the geometry of the earthquake faulting, **through** estimating the angular parameters strike, dip and rake"

Line 78: "*in time or in the frequency domain*"
I suggest to change to: "in **the** time or in the frequency domain"

Lines 85-86: Please, change "*affect*" to "**affects**" and "*the resolution errors refer to the capability*" to "the resolution errors **that are referred** to the capability"

Lines 90-92: "*In the case of focal mechanism, number of seismic stations, as well as seismic network geometry, and velocity structure of the crust influence the resolution and the reliability of the retrieved model*"
I suggest to change to: "In the case of focal mechanism, the number of seismic stations, as well as **the** seismic network geometry, and the velocity structure of the crust influence the resolution and the reliability of the retrieved model"

Line 99: "*In fact, its geometry may resolve*"
I suggest to change to: "In fact, **a given** geometry may resolve"

Line 102. I suggest to change "*features*" to "**constraints**"

Lines 103-104: "*So, the number of seismic stations, the size and geometry of network are defined after a preliminary phase based on the specific seismological target is evaluated*"
I suggest to change to: "So, the number of seismic stations, the size and geometry of network are defined after a preliminary phase based on **the evaluation of the specific seismological target**"

Line 107: Please, change "*is represented only by strongest earthquakes*" to " is represented only by **the** strongest earthquakes"

Line 110: I suggest to change "*we must increase the number of seismic stations for area unit building a dense seismic network*" to: "we must increase the number of seismic stations for area unit **by** building a dense seismic network"

Line 118. "synthetic data measurements" seems conflicting, measurement is used for real data.

Lines 245-246: "*As shown in Figure 2b, splitting the range of the data into equal-sized bins, we selected the focal mechanism corresponding to the most populated class*"
I suggest to change to: "As shown in Figure 2b, splitting the range of the data into equal-sized bins, we selected the focal mechanism corresponding **to the median value of** the most

populated class"

Line 248: I suggest to use: "Then, we decide**d** to test"

Line 252. I suggest to change "*and 3) those of the 2nd, 3rd, 4th bins*" to: "and 3) those of the 2nd, 3rd, 4th **most populated** bins"

Lines 277-278: "The median value of the distance of the farthest station is then used to select the seismic stations for which synthetic data are calculated."
Was this value used as a threshold value? Would the authors, please, specify.

Lines 349-350: "*On contrary, only for M1 focal mechanisms there is no improvement because the number of P-wave polarities is the same for both D2 and D3 datasets (Table 1)*"
D2 dataset only includes P/S spectral level ratios, I don't understand the sense of this sentence.

Lines 418-419: "*The methodology described in this work can be a valid tool to design or to test the performance of local seismic networks, operated to monitor natural or induced seismicity*"
I suggest to change to: "The methodology described in this work can be a valid tool to design or to test the performance of local seismic networks, **aimed at monitoring** natural or induced seismicity"

Lines 421-422: "*Although it is a theoretical study, many earthquake scenarios with several magnitude, locations and noise conditions can be simulated to mimic the real seismicity*"
It seems that the authors want to lessen the theoretical aspect of their research, but this does not make sense. The theoretical approach allows to explore the reliability of the estimates of FMs obtained from experimental data; this is well expressed in the paper.

Line 832: "*Figure 9. FME (strike, dip and dake error) maps*"
Typing error: **rake** instead of dake.

I don't always agree with the succession of figures. I would present figure 11 after figure 6, as well as figure 12 after figure 10. Furthermore, a table that summarizes the simulation parameters for each map could be useful for the reader:
Figure No., type of map, dataset, magnitude bin, depth,...

---

## Author Comment (AC1)

**Reply to Comment on se-2021-88**

**Anonymous Referee #1**

Referee comment on "A functional tool to explore the reliability of micro-earthquake focal mechanism solution for seismotectonic purposes" by Guido Maria Adinolfi et al., Solid Earth Discuss., https://doi.org/10.5194/se-2021-88-RC2, 2021

**General comments**

**R1:** *The work provides a valid tool for testing the reliability of seismic networks in determining fault plane solutions. This is profitable in seismotectonic studies and is strictly relevant to the goals of the Solid Earth special issue. The proposed method has been applied to microearthquakes recorded by the ISNet seismic network in Southern Italy in the period 2005-2011. The authors used a Bayesian approach that jointly inverts the P/S long-period spectral level ratios and the P polarities to infer the fault-plane solutions. They also describe an application to the ISNet catalogue. The work is well presented, the subdivision into paragraphs is appropriate, the methodology is clearly stated, as well as results and conclusions. I suggest minor revisions.*

**A:** We thank the referee for his/her detailed revision work. We greatly appreciated his/her suggestions and support. We followed the referee's advices to improve the quality of the manuscript.

**Specific comments**

**R1:** *Lines 267-269: "In order to overcome this limitation, we used an empirical relationship to define the number and the distance of the seismic stations that record a seismic signal as function of magnitude, once its epicentral location (grid node) and depth are fixed." If the authors used an empirical relationship, it should be given. Actually, later in the text they state that they chose for each magnitude bin the median values of the distance of the farthest triggered seismic station and of the number of P-wave polarities. Please, clarify this point.*

**A:** We agree with the Referee and we add one sentence in L314-317 in order to clarify the empirical approach. Moreover, we modified the "Method" (now "Methodology") paragraph adding the steps of the proposed methodology to clarify our analysis.

**R1:** *Lines 382-388: "Using the results of our simulations, we classified the focal mechanism provided by De Matteis et al., (2016) according to a quality code based on the resolution of fault kinematics (Table 2). In fact, we assigned to focal mechanisms of Irpinia instrumental seismicity a quality A, B and C for solutions corresponding to FM3, FM2 and FM1 kinematics, respectively. The quality A, B and C correspond to the average value of KA misfit (FM1=4.5° FM2=3.1°, FM3=2.4°) calculated for M1, M2 and M3 magnitudes using D3 dataset and considering earthquakes at 10 km depth with 5% gaussian errors." The authors indicated multiple factors that influence the goodness of the fault plane solution, such as the magnitude, the observables from waveforms, network geometry, the noise level. They should point out that this is an example of application of the method, and that the proposed classification only concerns the type of mechanism. I would suggest giving more emphasis to this paragraph.*

**A:** We thank the Referee and we greatly appreciate his/her suggestion. We added a sentence in the Conclusion paragraph in L479-480.

***Technical corrections***

**R1:** *Lines 29-30: "We applied this methodology, by computing synthetic data, to the local seismic network operated in the Campania-Lucania Apennines" I suggest changing to: "We applied this methodology, by computing synthetic data, to the local seismic network **operating** in the Campania-Lucania Apennines"*

*Line 46: "After the earthquake location, origin time and dimension are identified" I suggest changing to: "After the earthquake location, origin time and dimension **source** are identified"*

*Line 53: Please, change "become" to "**becomes**".*

*Line 61: Please, change "so much so that" to "**so much that**".*

*Lines 69-71: "These features are employed in a very simple way by several algorithms to constrain the geometry of the earthquake faulting estimating the angular parameters strike, dip and rake" I suggest changing to: "These features are employed in a very simple way by several algorithms to constrain the geometry of the earthquake faulting, **through** estimating the angular parameters strike, dip and rake"*

*Line 78: "in time or in the frequency domain" I suggest to change to: "in **the** time or in the frequency domain"*

*Lines 85-86: Please, change "affect" to "**affects**" and "the resolution errors refer to the capability" to "the resolution errors **that are referred** to the capability"*

*Lines 90-92: "In the case of focal mechanism, number of seismic stations, as well as seismic network geometry, and velocity structure of the crust influence the resolution and the reliability of the retrieved model" I suggest to change to: "In the case of focal mechanism, the number of seismic stations, as well as the seismic network geometry, and the velocity structure of the crust influence the resolution and the reliability of the retrieved model"*

*Line 99: "In fact, its geometry may resolve" I suggest to change to: "In fact, **a given** geometry may resolve"*

*Line 102. I suggest to change "features" to "**constraints**"*

*Lines 103-104: "So, the number of seismic stations, the size and geometry of network are defined after a preliminary phase based on the specific seismological target is evaluated" I suggest to change to: "So, the number of seismic stations, the size and geometry of network are defined after a preliminary phase based on **the evaluation of the specific seismological target**"*

*Line 107: Please, change "is represented only by strongest earthquakes" to " is represented only by **the** strongest earthquakes"*

*Line 110: I suggest to change "we must increase the number of seismic stations for area unit building a dense seismic network" to: "we must increase the number of seismic stations for area unit **by** building a dense seismic network"*

*Line 118. "synthetic data measurements" seems conflicting, measurement is used for real data.*

*Lines 245-246: "As shown in Figure 2b, splitting the range of the data into equal-sized bins, we selected the focal mechanism corresponding to the most populated class" I suggest to change to: "As shown in Figure 2b, splitting the range of the data into equal-sized bins, we selected the focal mechanism corresponding **to the median value** of the most populated class"*

*Line 248: I suggest to use: "Then, we decided to test"*

*Line 252. I suggest to change "and 3) those of the 2nd, 3rd, 4th bins" to: "and 3) those of the 2nd, 3rd, 4th **most** populated bins"*

**A:** Done.

**R1:** *Lines 277-278: "The median value of the distance of the farthest station is then used to select the seismic stations for which synthetic data are calculated." Was this value used as a threshold value? Would the authors, please, specify.*

**A:** We added a sentence (L314-317) to better clarify this point.

**R1:** *Lines 349-350: "On contrary, only for M1 focal mechanisms there is no improvement because the number of P-wave polarities is the same for both D2 and D3 datasets (Table 1)" D2 dataset only includes P/S spectral level ratios, I don't understand the sense of this sentence.*

**A:** We follow the referee's comment to clarify this point adding a sentence in L334-335. In our analysis, when D2 is simulated, in order to solve the verse ambiguity of the slip vector, a P-wave polarity is added to the earthquake data that will be inverted for the focal mechanism. As shown in Table1, for depth of 10 km, the number of P-wave polarities is 1. So, D2 and D3 datasets are the same for M1, with only one P-wave polarity.

**R1:** *Lines 418-419: "The methodology described in this work can be a valid tool to design or to test the performance of local seismic networks, operated to monitor natural or induced seismicity" I suggest to change to: "The methodology described in this work can be a valid tool to design or to test the performance of local seismic networks, **aimed at monitoring** natural or induced seismicity"*

*Lines 421-422: "Although it is a theoretical study, many earthquake scenarios with several magnitude, locations and noise conditions can be simulated to mimic the real seismicity" It seems that the authors want to lessen the theoretical aspect of their research, but this does not make sense. The theoretical approach allows to explore the reliability of the estimates of FMs obtained from experimental data; this is well expressed in the paper.*

*Line 832: "Figure 9. FME (strike, dip and dake error) maps" Typing error: **rake** instead of dake.*

**A:** Done. Thanks for the support.

**R1:** *I don't always agree with the succession of figures. I would present figure 11 after figure 6, as well as figure 12 after figure 10. Furthermore, a table that summarizes the simulation parameters for each map could be useful for the reader: Figure No., type of map, dataset, magnitude bin, depth,…*

**A:** We thank the Referee for his/her suggestion, but we prefer not to change the order of the figures. This order derives from a logic scheme that we followed in the main text to describe the analysis and the effects of the selected parameters. Additionally, as correctly suggested, we inserted the following table that summarizes the simulation parameters for each map and can facilitate the readability.

| Figure No. | Map | Focal Mechanism Solution | Magnitude Bin | Depth | Noise Level | Dataset |
|---|---|---|---|---|---|---|
| 4 | Kagan angle misfit | FM1, FM2, FM3 | M3 | 10 km | 5% | D1 |
| 5 | Kagan angle misfit | FM1 | M1, M2, M3 | 10 km | 5% | D2, D3 |
| 6 | Focal mechanism parameter misfit | FM1 | M1, M2, M3 | 10 km | 5% | D3 |
| 7 | Kagan angle average | FM1 | M1, M2, M3 | 10 km | 5% | D2, D3 |
| 8 | Kagan angle standard deviation | FM1 | M1, M2, M3 | 10 km | 5% | D2, D3 |
| 9 | Focal mechanism error | FM1 | M1, M2, M3 | 10 km | 5% | D3 |
| 10 | Kagan angle misfit | FM1, FM2, FM3 | M1, M2, M3 | 10 km | 5% | D3 |
| 11 | Focal mechanism parameter misfit | FM1 | M1, M2, M3 | 5 km | 5% | D3 |
| 12 | Kagan angle misfit | FM1 | M1, M2, M3 | 10 km | 30% | D3 |

**Table 2**

---

## Author Comment (AC2)

**Reply to Comment on se-2021-88**

*Anonymous Referee #2*

Referee comment on "A functional tool to explore the reliability of micro-earthquake focal mechanism solution for seismotectonic purposes" by Guido Maria Adinolfi et al., Solid Earth Discuss., https://doi.org/10.5194/se-2021-88-RC2, 2021

**R2:** *Manuscript by Guido Maria Adinolfi et al. attempts to provide methodology on how to explore the reliability of focal mechanisms inverted from events with generally small magnitudes. Using empirical data from Southern Italy they explore influence of different path- and network-geometry related parameters on stability of focal mechanism estimates.*

*From technical side, the manuscript by Guido Maria Adinolfi et al. is logically and well written. The introduction section providing general overview of moment tensor/focal mechanism inversion methodologies and related pitfalls could provide more complete overview of existing methods. The amplitude inversions are entirely missing, and the overview lacks of discussion of methods that accounts for some of the problems listed in the manuscript as factors contributing to inversion reliability. The methodology side discussing focal mechanism inversion and sampling/modelling/reliability assessment procedure could be in my opinion shortened, as vast majority of the text related to the focal mechanism inversion were already published in references manuscript. At the same time, the description of modelling procedure could be presented more clearly. The results and discussion parts are clear and to the point, as well as conclusion section (with major remarks regarding scientific content following below). Regarding the graphical presentation, the major issue I have is the use of colormaps that are not suitable for colorblind people. I made a reference in the detailed comments and ask the authors to replace the jet colormap with selected perceptually uniform one. Apart from the, the figures are clear and to the point.*

*As to the scientific content I have one major comment that I believed should be addressed somehow, or the Authors should consider reviewing the scope of the manuscript in the abstract and main text. The point is that the proposed parameters (and their range) tested in the simulation are very focused on the actual case study and they hardly allow to use this potentially interesting outcomes anywhere else in the other than conceptual way. As such, the manuscript presents the way the Authors dealt (nicely) with the reliability problem for THEIR case scenario, but in my opinion they failed a bit to provide a more general framework that could be actually useful for a broader audience. For example, the influence of magnitude on reliability of focal mechanisms is discussed by considering events with M1 M2 and M3. It is fine as such, but the magnitude is in fact a very indirect proxy influencing the quality of focal mechanism. The magnitude drives i.a. amplitude of ground motions including first pulse signal-to-noise ratio, but it is also a function of distance/depth and some other factors that ultimately drives whether the particular polarity is actually detected or not. So in my opinion, more general approach would be to prospect how the number of polarities that are well spread on the focal sphere (see next point of my argumentation) would affect the reliability of focal mechanism. Another example: The Authors present that outside of seismic network the reliability of focal mechanisms is decreasing dramatically. This is of course not a big surprise, but again Authors could in my opinion do better and more general here. Instead of barely presenting these outcomes in a form of maps showing strong deviations on edges of seismic network, they could actually test using their empirical data how the uniformity (i.e. azimuthal or takeoff-gap or both together!) of the focal sphere coverage affects the reliability of focal mechanism estimation. This could be achieved by cherry-picking the stations to enhance/limit the coverage of the focal sphere. My experience from the small scale seismicity is that the uniformity of coverage and number of stations are ultimately key*

*factors for reliable solutions. Using these factors would make the presented case more generic.*

*To summarize, I like overall the idea and empirical approach. However, I approached the manuscript with strong expectations of seeing a more generic approach to the problem of testing of reliability of focal mechanism. At the moment, the manuscript is more a case study, providing at the same time some conceptual framework, 'food for thoughts' as I would say, for the readers. I do believe the Authors could enhance its attractiveness/usability/generatily by discussing 'simulation' data using more generic range of parameters/variables, as proposed, but not limited to, to the ones discussed above.*

**A:** We thank the referee for his/her detailed and precise revision. We appreciate his/her suggestions and advice that help us to improve the manuscript. We accept his/her recommendation to modify the figures after selecting an appropriate colour scale with perceptually uniform colour scheme.

In this work we propose a tool that, despite our empirical approach, can be applied to different network configurations or to study a single focal mechanism solution. It is challenging to test the reliability of focal mechanism in a general framework if some parameters, as the network configuration or fault kinematic, are not fixed. For this reason, we applied our methodology to the Irpinia Seismic Network, and we carried out our analyses to study the reliability of focal mechanism solution for the local seismicity. We want to underline that we test the effects of different parameters as the magnitude, distance/depth of the earthquake or network geometry according to an empirical approach. As is well known, all these parameters influence the number of input data and the focal sphere coverage. We agree with the referee comment: "... *the magnitude is in fact a very indirect proxy influencing the quality of focal mechanism."* In fact, we take into account the magnitude effect on the amplitude and polarity of ground motion by varying for each magnitude bin (M1, M2 and M3) the number of input data (polarities and/or spectral level ratios) and the maximum source-receiver distance (see Table 1). The latters have been empirically estimated analysing the local seismicity recorded by ISNet. Moreover, for each magnitude and analysed hypocentre, the number and geometrical distribution of recording stations change and, consequently, the coverage of the focal sphere is modified. So, in this way, we test the effect of the coverage of the focal sphere and evaluate the reliability of the retrieved fault plane solutions.

Also, we want to underline that in the manuscript we propose a methodology that we applied to a specific scenario, but the obtained results allow us to make some considerations useful in a general framework. For example, with only six polarities data we cannot retrieve a stable and reliable focal mechanism solution for local earthquakes, even if a good azimuthal coverage is guaranteed (Figure 3). At the same time, if polarities and P/S amplitude spectral level ratios are used together, we are able to constrain the focal mechanism for local micro-seismicity with magnitude smaller than 3 (down to M 1.5) or to retrieve correctly fault plane solutions for earthquakes occurred close to the border (inside and outside) of the network. In addition, if only P/S spectral level ratio data are used, there are some cases inside the network for which, despite the optimal azimuthal coverage, the focal mechanism solution is not acceptable. These results suggest that specific analysis that takes into account the network geometry and, consequently, the coverage of the focal sphere, must be carried out.

In addition, we followed the referee's main comment and we carried out a specific test to explore the reliability of focal mechanism estimation in a more general framework. In order to explore the uniformity of focal sphere coverage, we simulated 10400 earthquakes fixing the fault plane solution and varying: 1) the number of seismic stations (6-30), 2) take-off angle and 3) azimuth of each single station. For each simulation, we computed the Kagan Angle (ka) between the theoretical and retrieved focal mechanism solutions, using only P-polarities. We show the results in the following figures. In the figures 1 and 2 the results are presented as 3-D histograms. In the figure 1, as expected, we note that, as the number of stations increases, the Kagan angle

and its range of variation decrease. Moreover, if the number of stations is less than nine, only few solutions have ka<40°. Figure 2 shows that the most value of ka less than 30° are obtained for azimuthal gap less than about 80°. In the figures 3, 4, 5, 6, we show the results as 3D scatter plot with the projections on the 3 coordinate planes. In these figures, it is evident the relation among the Kagan angle, azimuthal gap and number of stations. We added a short text describing the test and the Figures 1 and 2 in the manuscript, and the Figures 3-4-5-6 in the Supplementary Material.

[Figure]

**Figure 1.** 3D-histograms of the simulation results in terms of number of stations and Kagan angle misfit.

[Figure]

**Figure 2.** 3D-histograms of the simulation results in terms of azimuthal gap and Kagan angle misfit.

[Figure]

**Figure 3.** 3D-scatter plot of the simulation results in terms of number of stations (X-axis), azimuthal gap (Y-axis) and Kagan angle misfit (Z-axis).

[Figure]

**Figure 4.** Projection of the simulation results on the XY coordinate plane in terms of number of stations and azimuthal gap.

[Figure]

**Figure 5.** Projection of the simulation results on the ZY coordinate plane in terms of number of Kagan angle misfit and azimuthal gap.

[Figure]

**Figure 6.** Projection of the simulation results on the ZX coordinate plane in terms of Kagan angle misfit and number of stations.

*Detailed comments.*

**R2:** *P2 L64-66. I note here there is missing overview paper by Bentz at al. [1] that discusses peculiarities of moment tensor inversion for small events while using full waveform and amplitude approaches.*

*P2 L64-66. I would update paper with some newer references for the full-waveform inversion, for example Kiwi tools is missing, to name a few.*

*P3 L67-83. This part tackles problem of focal mechanism inversion from very general perspective. However, the description is at the moment not complete and must be updated. For example, P-wave amplitude inversion methods which are very popular in the magnitude range of concern in this study are not mentioned. Example of methods and software packages include PCA approach [2], source-function deconvolution (papers from Jan Sileny et al. from Prague). Moreover, as this and later part of introduction discuss external factors influencing quality of MT/focal mechanism inversion, there are already existing methods addressing these problems that could be referred to as well (e.g. [3], to name a few!). Please provide up-to-date description of the state-of-the-art in the field then.*

**A:** We agree with the Referee and we modified the text adding a specific part (L65-73) with references about the new methods proposed for the moment tensor inversion of small events.

**R2:** *P3 L109-111. PCA approach is good solution here and should be mentioned [2] in my opinion.*

**A:** We added the reference in L65-70 in which the moment tensor inversion is discussed. We followed the previous Referee's suggestion.

**R2:** *Introduction part as a whole: You are not mentioning explicitly the potential biases related to double-couple model assumption. Many small earthquakes are in fact not following*

*double-couple especially in case of induced seismicity (that you incorporate in your manuscript in discussion). I believe this potential modelling bias should be mentioned as another source of errors with appropriate referencing.*

**A:** We agree with the Referee and we modified the manuscript adding some sentences about this modelling bias.

**R2:** *L101-113 In general, some of your parameters can be consider a "quality of coverage of hypocenter with seismic stations with sufficient signal-to-noise ratio". I feel this part is a bit too much of the text overall.*

**A:** We follow the Referee's advice and we deleted some sentences of the main text.

*Method:*

**R2:** *L131-133. Any references? What magnitude range? This overall sounds to me a bit bold statements, or maybe that refers to larger magnitude events (?). My experience that is largely related to analysis of focal mechanisms of very small events is that S/P amplitude ratios are actually very hard to constrain manually.*

**A:** We follow the Referee's advice and we modified the manuscript adding a reference. In De Matteis et al. (2016), P/S amplitude ratios have been used also to constrain the focal mechanism solutions of small earthquakes (M<3). We agree with the Referee's comment about the complication of the estimate of P/S amplitude ratios, especially for small earthquakes in time domain. Nevertheless, we think that inverting the P- and S-wave displacement Fourier spectra allow a robust low frequency level estimates when the signal to noise ratio is acceptable. Moreover, the measure of P/S amplitude ratios can be automated in the signal processing.

**R2:** *L140 and following. Do we really need this very detailed description of BISTROP code? I think you could easily skip the very first part by referring to key input and outputs of the code and focus more on last part, which is typically less familiar to the reader. It is quite unclear actually how the bayesian framework was use with respect to quality criteria parameters you used (kagan angle and others).*

**A:** We agree with the Referee and we modified the manuscript deleting some sentences.

**R2:** *L264-267. I agree empirical approach pursued here is better, but in general case approaches could be used. See [4-5] and some nice references therein that cover problems of detection using empirical data as well.*

**A:** We added the suggested references.

**R2:** *L278. I believe the "synthetic" word might cause a confusion, as it suggests that you generate or model something from scratch. In fact, as I understand you resample your empirical polarity data to create "new" events taking into account different constraints posed beforehand, at least this is how I understand it. All in all, maybe revise this part to make it more understandable (bullet points? scheme? chart?)*

**A:** We agree with the Referee and we modified the "*Method*" paragraph (now "*Methodology*") adding the scheme of earthquake data simulations to clarify our analyses. We did not carry out "a resampling of the empirical data", nor a sort of bootstrap. We simulated earthquake data (P-wave polarities or P/S spectral level ratios) for each grid node (with different magnitude and

noise level) taking into account different constraints derived from ISNet data (Table 1) in terms of number and maximum distance of seismic stations for which synthetic data were computed. For this reason, we think that the term "synthetic" is appropriate, but we are open to other Referee or Editor suggestions, if necessary.

**R2:** *L296. It is actually not clear (or I missed that) where this Gaussian noise is applied (to the P/S ratios?), and if the latter is the case, how it is actually applied (formula?)*

**A:** We added zero mean Gaussian noise to P/S spectral level ratios with a standard deviation equal to two different percentage levels, as 5% and 30%. We modified L324-325 and clarified where Gaussian noise is applied in the scheme of earthquake simulation that we have inserted in L150-154. Please, see our previous reply and "*Methodology*".

**R2:** *Figure 1. No axes labels!*

*Figure 3. Maybe some scale just to give impression how big the area is. Axis labels missing, e.g. "grid points along latitude"*

*Figures 4,5,6... Axes' labels missing. As you want to optimize the space (which is fine), why don't you just mention them in caption what is the horizontal and vertical direction. Figures 4,5,6... please DO replace 'jet' colormap with ANY color-blind friendly colormap in all plots. For a general guidance, this paper is a treasure: https://www.nature.com/articles/s41467-020-19160-7*

**A:** Done!

**R2:** *L335 Kagan angle cannot be negative (?)*

**A:** We agree with the Referee and for this reason we clarify the point in L376-377. We assume KA = -1 as an indeterminate value; this is valid for the nodes of the grid where it is not possible to retrieve the solution of the focal mechanism. For these nodes the minimum number of stations (at least 6) are not available. Assuming a KA=-1 allow us to report this information in the color bar (white triangle) used in the figures.

**R2:** *L378 Typo*

*L377 'is less' or 'is smaller' ?*

**A:** Done!

References
[1]Bentz, Stephan, P. Martínez-Garzón, G. Kwiatek, M. Bohnhoff, and J. Renner (2018). Sensitivity of Full Moment Tensors to Data Preprocessing and Inversion Parameters: A Case Study from the Salton Sea Geothermal Field. Bull. Seismol. Soc. Am. 108, 588–603, doi 10.1785/0120170203

[2]Vavrycuk, V., P. Adamova, J. Doubravová, and H. Jakoubková (2017). Moment Tensor Inversion Based on the Principal Component Analysis of Waveforms: Method and Application to Microearthquakes in West Bohemia, Czech Republic. Seismological Research Letters 88, 1303–1315, doi 10.1785/0220170027

*[3]Kwiatek, G., P. Martínez-Garzón, and M. Bohnhoff (2016). HybridMT: A MATLAB Software Package for Seismic Moment Tensor Inversion and Refinement. Seismol. Res. Lett.*

*[4]Kwiatek, G. and Y. Ben-Zion (2020). Detection Limits and Near-Field Ground Motions of Fast and Slow Earthquakes. Journal of Geophysical Research: Solid Earth 125, e2019JB018935, doi 10.1029/2019JB018935*

*[5]Kwiatek, G. and Y. Ben-Zion (2016). Theoretical limits on detection and analysis of small earthquakes. Journal of Geophysical Research-Solid Earth 121, doi 10.1002/2016JB012908*

*[6]Stierle, E., M. Bohnhoff, and V. VavryÄ•uk (2014). Resolution of non-double-couple components in the seismic moment tensor using regional networks—II: application to aftershocks of the 1999 Mw 7.4 Izmit earthquake. Geophys. J. Int. 196, 1878–1888, doi 10.1093/gji/ggt503*

*[7]Stierle, E., V. VavryÄ•uk, J. šílený, and M. Bohnhoff (2014). Resolution of non-doublecouple components in the seismic moment tensor using regional networks—I: a synthetic case study. Geophys. J. Int. 196, 1869–1877, doi 10.1093/gji/ggt502*

---

## Author Response (AR1)

**SE-2021-88**

*Reply to Editor's review of the article:*
*A functional tool to explore the reliability of micro-earthquake focal mechanism solution*
*Guido Maria Adinolfi, Raffaella De Matteis, Rita De Nardis, and Aldo Zollo*

Dear Editor

thank you for your suggestions and advice on the colour scale of the figures. We have accepted your decision, and we have modified all the figures (and their panels) with an appropriate colour scale with perceptually uniform scheme.

We have carried out a detailed and thorough review of the manuscript. For that, we thank the Referees for their valuable advices. We have followed their suggestions and have modified the main text by improving its readability and adding a few sentences to clarify our work.

In particular, we have modified the layout of the "Methodology" paragraph according to a scheme, as suggested by both Referees, and we have organized the "Data Analysis" with the same structure.

In addition, we have followed the 2nd Referee's main comment and we have carried out a specific (totally new) test to explore the reliability of focal mechanism estimation in a more general framework. In order to explore the uniformity of focal sphere coverage, we simulated 10400 earthquakes fixing the fault plane solution and varying: 1) the number of seismic stations (6-30), 2) take-off angle and 3) azimuth of each single station. We have added a short text describing the test and one figure in the main text and another in the Supplementary Material.

Finally, we have carefully revised the English language.

We hope that the new version of the manuscript, with the many improvements derived from your suggestions as well as from the Referees, will be deemed suitable for publication on Solid Earth.

Best regards,

Guido Maria Adinolfi

---

## Author Response (AR2)

**SE-2021-88**

***Reply to Editor's review of the article:***
***A functional tool to explore the reliability of micro-earthquake focal mechanism solution***
*Guido Maria Adinolfi, Raffaella De Matteis, Rita De Nardis, and Aldo Zollo*

Dear Editor

thank you for your detailed corrections. We have accepted all your suggestions.

In the meantime, we are modifying the reference list according to the SE standards.

Best regards,

Guido Maria Adinolfi

---

## Author Response (AR3)

**SE-2021-88**

***Reply to Editor's review of the article:***
***A functional tool to explore the reliability of micro-earthquake focal mechanism solution***
*Guido Maria Adinolfi, Raffaella De Matteis, Rita De Nardis, and Aldo Zollo*

Dear Editor

thank you for your detailed corrections. We have accepted all your suggestions.

We modified the reference list according to the SE standards.

Best regards,

Guido Maria Adinolfi